# Molecular structures of the eukaryotic retinal importer ABCA4

**Fangyu Liu**[1,2†]**, James Lee**[1]**, Jue Chen**[1,3]*

[1]Laboratory of Membrane Biology and Biophysics, The Rockefeller University, New York, United States; [2]Tri-Institutional Training Program in Chemical Biology, New York, United States; [3]Howard Hughes Medical Institute, Chevy Chase, United States

**Abstract** The ATP-binding cassette (ABC) transporter family contains thousands of members with diverse functions. Movement of the substrate, powered by ATP hydrolysis, can be outward (export) or inward (import). ABCA4 is a eukaryotic importer transporting retinal to the cytosol to enter the visual cycle. It also removes toxic retinoids from the disc lumen. Mutations in ABCA4 cause impaired vision or blindness. Despite decades of clinical, biochemical, and animal model studies, the molecular mechanism of ABCA4 is unknown. Here, we report the structures of human ABCA4 in two conformations. In the absence of ATP, ABCA4 adopts an outward-facing conformation, poised to recruit substrate. The presence of ATP induces large conformational changes that could lead to substrate release. These structures provide a molecular basis to understand many disease-causing mutations and a rational guide for new experiments to uncover how ABCA4 recruits, flips, and releases retinoids.

*For correspondence:
juechen@rockefeller.edu

Present address: †Department of Pharmaceutical Chemistry, University of California, San Francisco, San Francisco, United States

**Competing interests:** The authors declare that no competing interests exist.

## Introduction

ABCA4, also known as the Rim protein or ABCR, is an ATP-binding cassette (ABC) transporter essential to vision. Over 800 mutations in the ABCA4 gene cause Stargardt disease, the most common form of inherited macular degeneration (*Lin et al., 2016*; *Ran et al., 2014*). Several other diseases, including age-related macular degeneration, autosomal-recessive retinitis pigmentosa, and cone-rod dystrophy, are also linked to defective ABCA4 (*Allikmets et al., 1997*). Currently, no cure nor effective treatment exists for these diseases. Understanding the structure and function of ABCA4 is important to uncovering the underlying mechanisms and facilitating therapy development.

The molecular properties of ABCA4 are unique and intriguing. It belongs to the large family of ABC transporters, which include thousands of ATP-powered pumps translocating different substrates across the membrane (*Molday, 2015*). Prokaryotic ABC transporters include both importers that bring substrates into the cytosol and exporters that move substrates out of the cytosol. In eukaryotes, however, the vast majority of ABC transporters are exporters. ABCA4 is an exception; it functions as an importer in photoreceptor cells, where light stimuli are translated into electric signals (*Figure 1A*; *Illing et al., 1997*; *Papermaster et al., 1982*). In the dark, 11-*cis* retinal binds the inert opsin. Light catalyzes the isomerization of 11-*cis* retinal to all-*trans* retinal, which is subsequently released from the activated opsin. For photoreceptor cells to function continuously, all-*trans* retinal must be converted back to 11-*cis* retinal through a series of enzymatic reactions known as the visual cycle (*Figure 1A*). ABCA4 enables this process by moving all-*trans* retinal into the cytoplasm, likely in the form of its phosphatidylethanolamine (PE) conjugates, N-retinylidene-PE (*Figure 1A*). In addition, ABCA4 also clears excess *trans* and 11-*cis* retinal out of the disc lumen to prevent potential toxicity (*Quazi and Molday, 2014*; *Weng et al., 1999*).

Retinoids have been shown to bind to purified ABCA4 and stimulate its ATPase activity (*Ahn et al., 2000*; *Sun et al., 1999*). Transporter assays have demonstrated that ABCA4 actively transports N-retinylidene-PE across disc membranes (*Quazi et al., 2012*; *Quazi and Molday, 2014*).

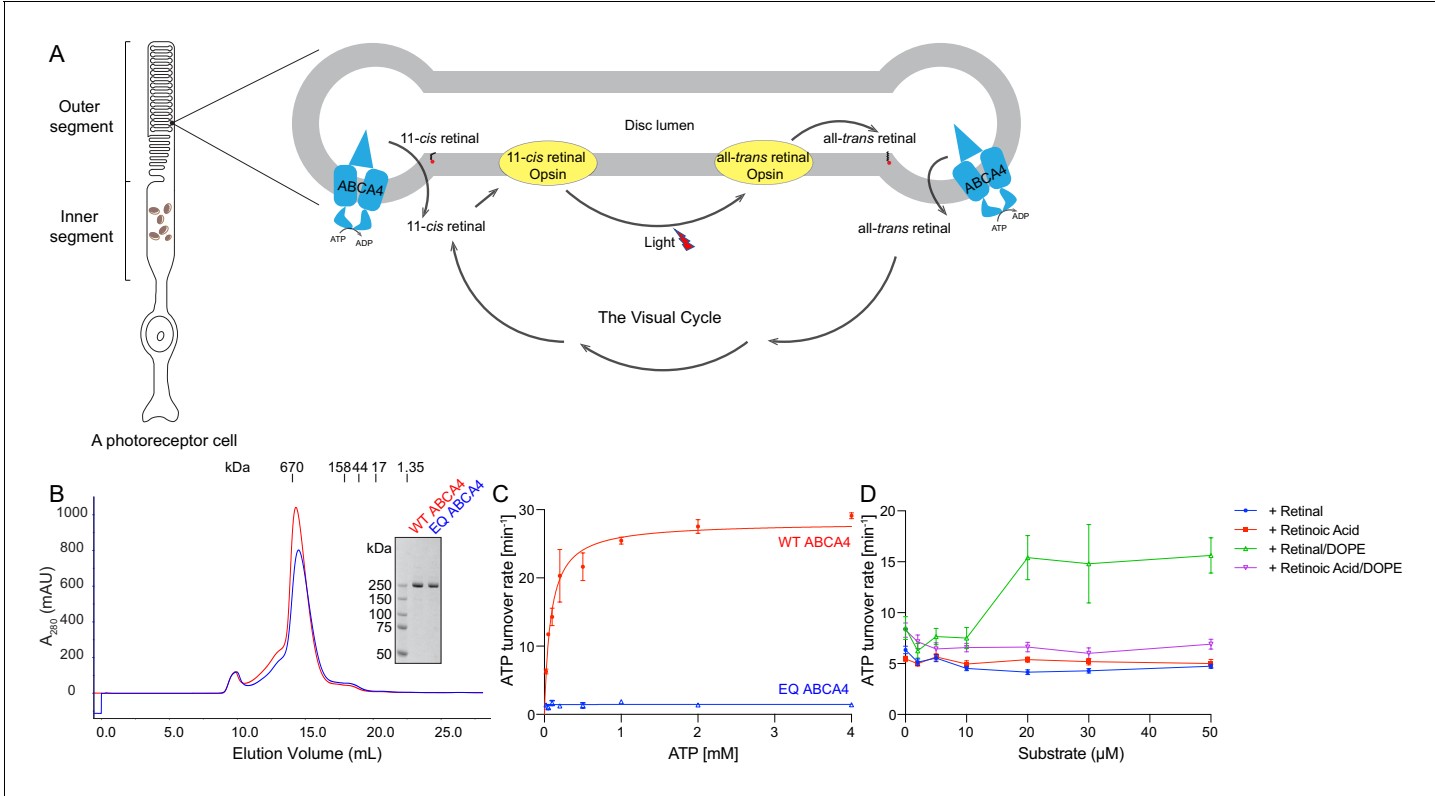

**Figure 1.** Biochemical characterization of ABCA4. (**A**) An illustration showing a rod photoreceptor cell and a zoomed-in view of the outer segment disc, where ABCA4 is located. ATP-hydrolysis enables ABCA4 to transport all-*trans* retinal and 11-*cis* retinal from the disc lumen into cytosol. (**B**) Size exclusion profile of purified wild-type (WT) ABCA4 and the E1087Q/E2096Q mutant (EQ). (**C**) Basal ATPase activity measured in 0.06% digitonin at 28°C. Data points represent the means and standard deviations (SDs) of three measurements. The WT ABCA4 has a $K_m$ of 0.08 ± 0.01 mM and specific turnover rate of 28.1 ± 0.7 ATP per minute, corresponding to a maximal ATPase activity of 112.5 ± 2.8 nmol/mg/min. (**D**) The ATPase activity of purified ABCA4 as a function of all-*trans* retinal or all-*trans* retinoic acid in the presence or absence of 0.1 mg/mL 1,2-dioleoyl-sn-glycero-3-phosphoethanolamine (DOPE) and 100 µM ATP. Data points represent the means and SDs of three measurements from the same protein preparation.

These studies provide direct evidence that ABCA4 is an ATP-driven importer, transporting N-retinylidene-PE from the luminal to the cytoplasmic side of the disc membrane in photoreceptor cells.

Although significant progress was made in understanding the physiological role of ABCA4, structural information is limited. Previously, the overall architecture of bovine ABCA4 was characterized by electron microscopy (EM) to 18 Å resolution (*Tsybovsky et al., 2013*). More recently, the structure of a related transporter, ABCA1, was determined to be 4.1 Å (*Qian et al., 2017*). ABCA1 is a lipid exporter sharing 51% sequence identity with ABCA4 but transports substrate in the opposite direction.

In this study, we used cryo-electron microscopy (cryo-EM) to determine the structures of human ABCA4 in two conformations. These structures, both determined at 3.3 Å resolution, reveal ATP-dependent conformational changes different from those observed in any other ABC transporters.

## Results

### Recombinant ABCA4 functions similarly to protein purified from native source

The wild-type (WT) ABCA4 and a double-mutant E1087Q/E2096Q (EQ) were overexpressed in mammalian cells and purified to homogeneity (*Figure 1B*). The E1087 and E2096 residues were conserved as the general bases catalyzing ATP hydrolysis. Mutation of the corresponding residues in many other ABC transporters abolished ATP hydrolysis but not ATP binding (*Kim and Chen, 2018*; *Zhang et al., 2017*; *Zhang et al., 2018*). The size-exclusion profiles of the purified EQ mutant and

the WT protein are comparable, suggesting that the EQ mutant folds similarly to the WT protein (*Figure 1B*).

The ATPase activity of ABCA4 was measured in detergent solution as a function of ATP concentration (*Figure 1C*). The recombinant WT protein shows a $K_m$ of 80 ± 10 µM for ATP and a maximal hydrolysis activity of 28.1 ± 0.7 ATP per minute (or 112.5 ± 2.8 nmol/mg/min), comparable to those of the native protein purified from rod photoreceptor cells (75 ± 18 µM and 190 ± 36 nmol/mg/min, respectively) (*Ahn et al., 2000*). The EQ mutant, on the other hand, shows no measurable ATPase activity (*Figure 1C*). Also consistent with literature (*Quazi and Molday, 2014*), in the presence of PE, retinal stimulates the ATPase activity of WT ABCA4 by up to threefold (*Figure 1D*). No stimulation was observed with retinoic acid or in the absence of PE (*Figure 1D*). These data indicate that the recombinant ABCA4 exhibits biochemical properties very similar to the ABCA4 purified from native sources.

## ABCA4 adopts an outward-facing conformation in the absence of ATP

The human ABCA4 is a single polypeptide of 2273 residues (*Figure 2*). It consists of two homologous halves, each containing a transmembrane domain (TMD), an exocytoplasmic domain (ECD), a cytoplasmic nucleotide binding domain (NBD), and a regulatory domain (RD) (*Bungert et al., 2001*). The structure of WT ABCA4 was determined by single-molecule cryo-EM in the absence of ATP and substrate to an overall resolution of 3.3 Å (*Figure 2—figure supplement 1*, *Table 1*). The EM density is of high quality (*Figure 2—figure supplement 2*), permitting de novo building of most residues, except for a few loops and a 134-residue region in ECD1 (*Figure 2B*, blue density). In addition, residues 118–137 and 479–496 were built as polyalanine as their side chain densities were invisible. The final model, consisting of 1941 residues, 14 lipids, 1 detergent molecule, and 14 sugar molecules, was refined to excellent stereochemistry (*Table 1*).

ABCA4 is an elongated molecule, extending approximately 240 Å across the lipid bilayer (*Figure 2B*). The two ECDs pack closely with each other, protruding 130 Å into the lumen (*Figure 2B*). The two TMDs adopt an outward-facing configuration, forming a large hydrophobic cavity that is continuous with both the luminal solution and, through a lateral opening, to the lipid bilayer itself. In the absence of ATP, the two NBDs are apart from each other and the two RDs form extensive interdomain interactions (*Figure 2B*). It was previously suggested that the highly specific localization of ABCA4 to the rim region of the disc membrane (*Figure 1A*) might be due to the large size of the ECD domains (*Molday, 2015*). The structure of ABCA4 supports this notion as the spacing between the flattened regions of the disc membranes (*Nickell et al., 2007*) may be insufficient to accommodate the elongated ECDs.

## An extended groove of the ECDs

The luminal region, composed of 600 residues in ECD1 and 290 residues in ECD2, does not contain any known enzymatic or structural motifs of known function. However, more than 300 disease-causing mutations map to this region (*Figure 3—figure supplement 1*) (Human Gene Mutations Database: http://www.hgmd.cf.ac.uk), underscoring the essential nature of the ECDs.

In contrast to the rest of the molecule, where the N- and C-terminal halves have similar structures, pseudo twofold symmetry does not exist in the ECDs. Instead, the ECDs assemble to form one unit with a large extended hollow interior, accessible from the luminal space (*Figure 3*). The structure, as previously described for ABCA1 (*Qian et al., 2017*), can be separated into three layers: base, tunnel, and lid (*Figure 3A*). The base has a global fold docked onto the luminal surface of the two TMDs. The tunnel is an -helical barrel with an inner diameter of approximately 15 Å (*Figure 3A, D*). The density corresponding to the lid region does not permit amino acid assignment, suggesting that the lid region is flexible (*Figure 3A, C*). Nevertheless, it is clear that the interior of the lid is also hollow (*Figure 3C*). Strong densities that may correspond to lipid or detergent molecules were observed inside the extended groove in the base layer (*Figure 3A, B*, *Figure 2—figure supplement 2*).

Consistent with biochemical analysis (*Bungert et al., 2001*), the structure of the ECDs is stabilized by six disulfide bonds (*Figure 3E*). Seven missense mutations of the cysteines involved in disulfide bonding (C54Y, C75G, C519R, C641S, C1455R, C1488R, and C1490Y) have been identified in patients with retinal diseases (*Fumagalli et al., 2001*; *Hu et al., 2019*; *Lewis et al., 1999*; *Rosenberg et al., 2007*). Previously, mass spectrometry (*Tsybovsky et al., 2011*) and mutagenesis

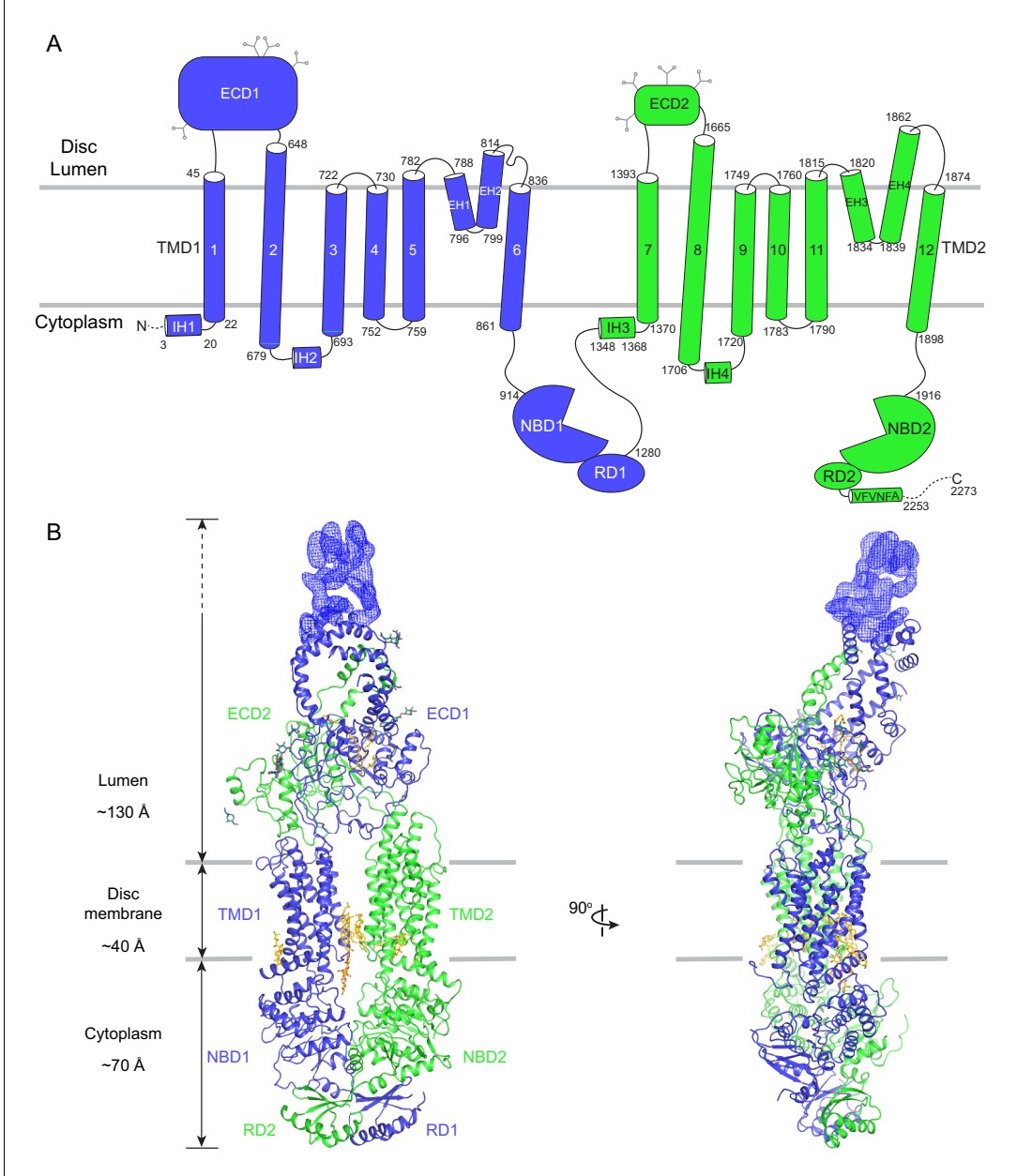

**Figure 2.** The overall structure of ABCA4 in the absence of ATP. (**A**) The domain structure of ABCA4. The N- and C-terminal halves of the molecule are shown in blue and green. (**B**) Two orthogonal views of ABCA4 in ribbon presentation. Also shown is the electron microscopy density corresponding to residues 138–271 in ECD1. Ordered detergents, lipids, and N-linked glycans are shown as stick model. The position of the membrane is indicated by two gray lines.

The online version of this article includes the following figure supplement(s) for figure 2:

**Figure supplement 1.** Cryo-electron microscopy reconstructions of the ATP-free, wild-type ABCA4.

**Figure supplement 2.** Local density of the ATP-free, wild-type ABCA4 reconstruction.

studies (*Bungert et al., 2001*) have identified multiple N-linked glycosylation sites in the ECDs. Densities corresponding to sugar moieties were observed on residues N98, N415, N444, N504, N1469, N1529, N1588, and N1662. It is likely that glycosylation plays an important role in the folding or trafficking of ABCA4,as missense mutations at positions 415, 504, and 1588 are linked to retinal diseases (*Consugar et al., 2015*; *Khan et al., 2019*; *Kim et al., 2019*).

**Table 1.** Summary of electron microscopy data and structure refinement statistics for wild-type ABCA4.

| *Data collection* | |
|---|---|
| Microscope | Titan Krios (FEI) |
| Voltage (kV) | 300 |
| Detector | K2 Summit (Gatan) |
| Pixel size (Å) | 1.03 |
| Defocus range (µm) | 0.8–2.5 |
| Movies | 5226 |
| Frames/movie | 50 |
| DDose rate (electrons/pixel/s) | 8 |
| Total dose (electrons/Å$^2$) | 75 |
| Number of particles | 1,375,284 |
| *Model composition* | |
| Non-hydrogen atoms | 15,789 |
| Protein residues | 1941 |
| Lipids | 14 |
| Digitonin | 1 |
| Sugar molecules | 14 |
| *Refinement* | |
| Resolution (Å) | 3.3 |
| Sharpening B-factor (Å$^2$) | −99.8 |
| Root-mean-square deviations | |
| Bond lengths (Å) | 0.006 |
| Bond angles (°) | 1.185 |
| *Validation* | |
| Molprobity score | 2.16 |
| Clashscore, all atoms | 18.86 |
| Favored rotamers (%) | 97.10 |
| Ramachandran plot (%) | |
| Favored | 94.28 |
| Allowed | 5.72 |
| Outliers | 0.0 |

Prokaryotic ABC importers often contain an extracytoplasmic binding protein that functions as a receptor for substrates (*Davidson et al., 2008*). Currently it is unknown if the ECDs of ABCA4 have an analogous function. Previously it was shown that isolated ECD2 binds to all-*trans* retinal (*Biswas-Fiss et al., 2010*), but how this interaction relates to the transport cycle remains unclear.

## The transmembrane region contains a hydrophilic cleft

The structure of the TMD belongs to the newly classified type V fold (*Thomas and Tampé, 2020*) previously observed only in ABC exporters (*Lee et al., 2016*; *Qian et al., 2017*; *Taylor et al., 2017*; *Bi et al., 2018*). The two TMDs come in contact with each other through two helical turns at the cytoplasmic end of TM5 and TM11, burying only 124 $^2$ surface per subunit (*Figure 4A*). The gap between the TMDs at the level of the inner leaflet is filled with ordered lipids (*Figure 4A, B*). Thus, the cytoplasmic gate of the transmembrane pathway is formed partially by residues in TM5 and TM11 and partially by lipid molecules. Mutations of many lipid-interacting residues are found in patients with vision diseases (Human Gene Mutations Database: http://www.hgmd.cf.ac.uk). These

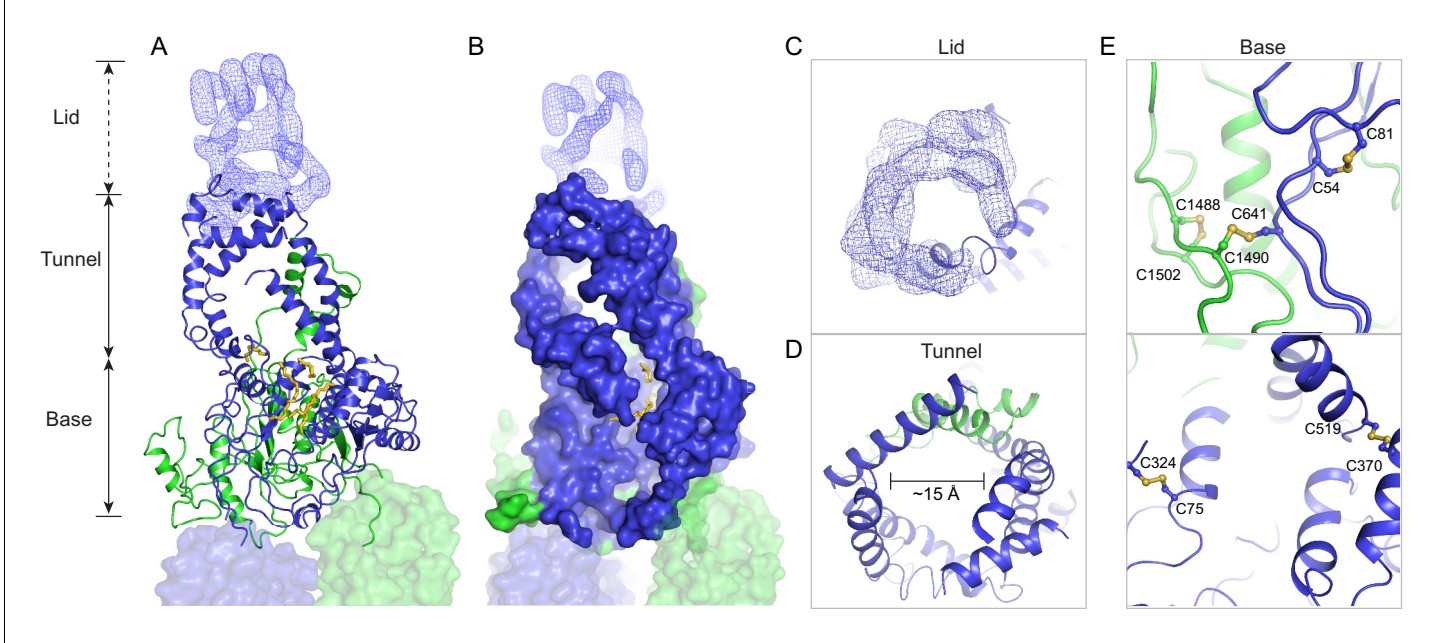

**Figure 3.** A three-tiered structure of the exocytoplasmic domains (ECDs). (A) Ribbon and (B) surface representation of the ECDs, together with the electron microscopy (EM) density of the lid region. Bound detergents and lipids are shown as yellow sticks. (C) A luminal view of the lid; the EM density is shown as blue mesh. (D) A cross section of the tunnel region. (E) The ECDs are stabilized by inter- and intra-domain disulfide bonds. The online version of this article includes the following figure supplement(s) for figure 3:

**Figure supplement 1.** Disease-causing mutations (orange) are widely distributed throughout the structure.

observations indicate that lipids form an integral part of the retinoid transport system, possibly by regulating the folding and function of ABCA4 (*Figure 1*, *Ahn et al., 2000*).

A prominent feature of the type V fold is that each TMD contains two exocytoplasmic helices (EHs) forming a EH-turn-EH insertion halfway into the membrane (*Figure 4B, C*). Each EH exposes a helical end to the low dielectric membrane center, a thermodynamically unfavorable configuration. In ABCA4, this configuration is stabilized by an acidic residue and a tyrosine residue on the neighboring TM helices (*Figure 4B, C*). In TMD1, D846 and Y850 form hydrogen bonds with mainchain carbonyl oxygen and amide atoms of EH1 and EH2, respectively (*Figure 4B*, right panel). In TMD2, E1885 and Y1889 neutralize the ionizable atoms at the ends of EH3 and EH4 (*Figure 4C*, right panel). Mutating E1885 to lysine causes Stargardt disease (*Rivera et al., 2000*), likely due to folding defects. The EH3-turn-EH4 insertion leads to a large hydrophilic cleft on the surface of TMD2 (*Figure 4C*). A cluster of charged residues from EH3, EH4, and TM12, including the clinically relevant residue R1843 (*Lewis et al., 1999*; *Rivera et al., 2000*), are located as deep as 10 below the expected membrane surface (*Figure 4C*).

In some $K^+$ or $Cl^-$ channels, helices with one end exposed to the membrane contribute to the binding and selection of conducting ions (*Doyle, 1998*; *Faraldo-Gómez et al., 2004*). Although the EH-turn-EH insertion is highly conserved in type V transporters, the functional significance of this motif is still unknown.

## The RDs exhibit ACT-like folds

In the cytoplasm, the two NBDs and two RDs form a domain-swapped dimer (*Figure 2B*). RD1 crosses the dimer interface to interact with NBD2, and RD2 crosses back over to pack against NBD1. The two RDs have very similar structures; both exhibit a fold similar to the ACT domain (*Figure 4D*), which is a conserved structural motif that typically binds small regulatory molecules (*Chipman and Shaanan, 2001*; *Grant, 2006*). Sequence alignment indicates that the ACT fold is conserved among all members of the ABCA subfamily (*Figure 4—figure supplement 1*).

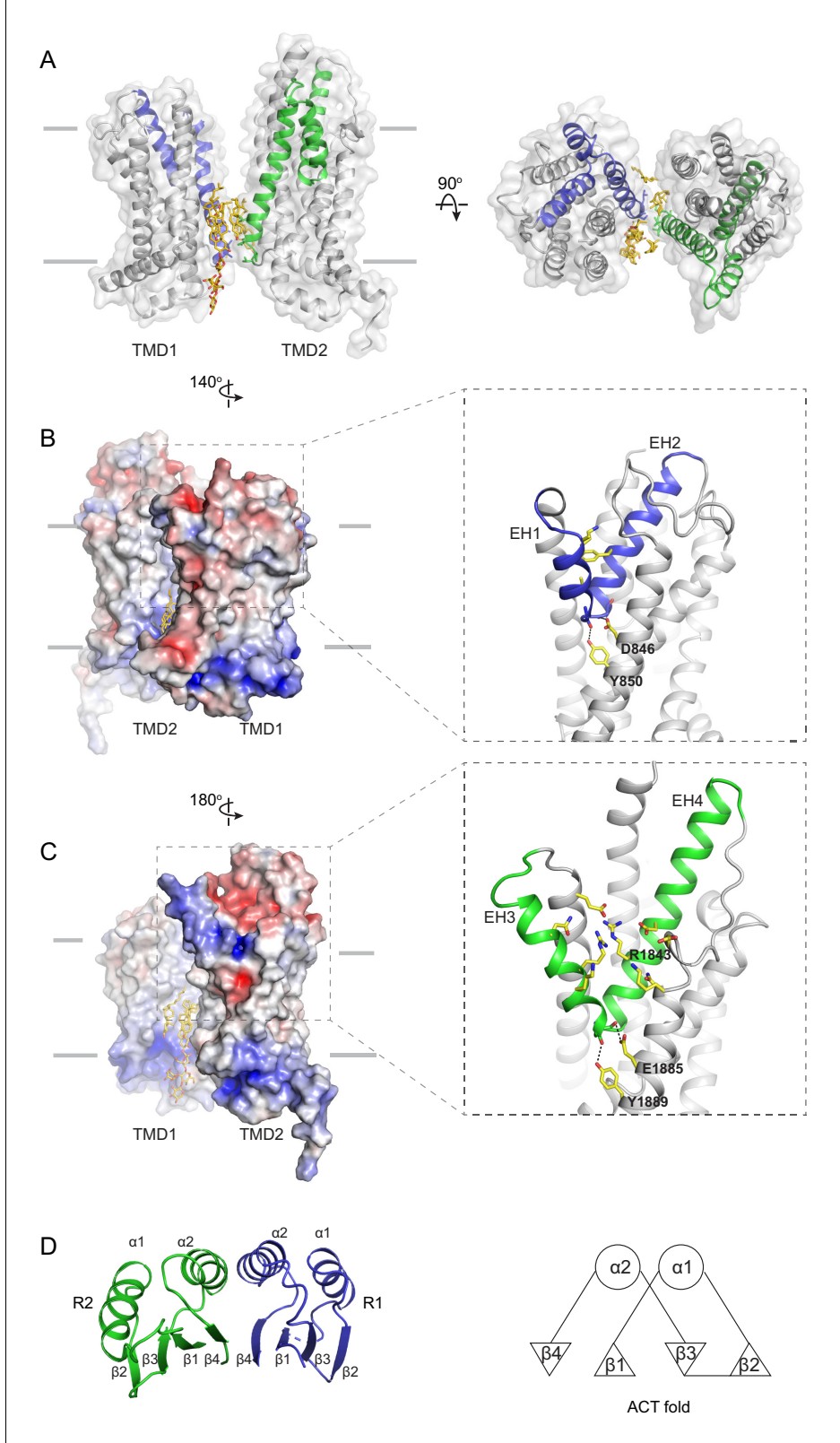

**Figure 4.** The structures of the transmembrane domains (TMDs) and regulatory domains (RDs). (**A**) Two views of the TMDs. EH1, EH2, and TM5 are highlighted in blue, and EH3, EH4, and TM11 are in green. The side chain of residues making inter-subunit contacts is shown as stick model. Lipids are shown in yellow. (**B**) Electrostatic surface representation of the TMDs, calculated assuming a pH of 7 and a concentration of 0.15 M of both (+1) and (−1)

*Figure 4 continued on next page*

*Figure 4 continued*

ions. Scale: red, negative (−5 kT/e); blue, positive (+5 kT/e). The EH1-turn-EH2 insertion is highlighted as blue ribbon. (C) An orthogonal view of (B), highlighting the hydrophilic indentation formed by the EH3-turn-EH4 insertion. (D) The RDs exhibit the ACT-like fold with the $\beta\alpha\beta\beta\alpha\beta$ topology. EH: exocytoplasmic helices.

The online version of this article includes the following figure supplement(s) for figure 4:

**Figure supplement 1.** The ACT motifs and the pinning helices (PHs) are conserved in the ABCA subfamily.

In the ABC transporter superfamily, the bacterial methionine importer MetNI also contains two RDs with ACT-like folds. The RDs of MetNI play an important role in 'trans-inhibition': at high intracellular concentration, methionine binds to the RDs to inhibit further uptake of methionine (*Kadaba et al., 2008*). For ABCA4, it was reported that an NBD1-R1 construct binds retinal with high affinity (*Biswas-Fiss et al., 2012*). Whether retinal binding is mediated through the ACT motif and the physiological relevance of such interaction remain to be investigated.

## Structure of ABCA4 in the presence of ATP

To capture ABCA4 in its ATP-bound state, the structure of the hydrolysis-deficient mutant ABCA4-EQ was determined in the presence of 10 mM ATP-Mg$^{2+}$ to 3.3 Å resolution (*Figure 5*, *Figure 5— figure supplements 1* and *2*, *Table 2*). Both 2D and 3D image classes from cryo-EM data show that most particles are in the NBD-dimerized configuration. The final model contains 2 ATP molecules, 8 lipid molecules, and 1920 residues. Similar to that of the WT protein, the lid region is highly mobile.

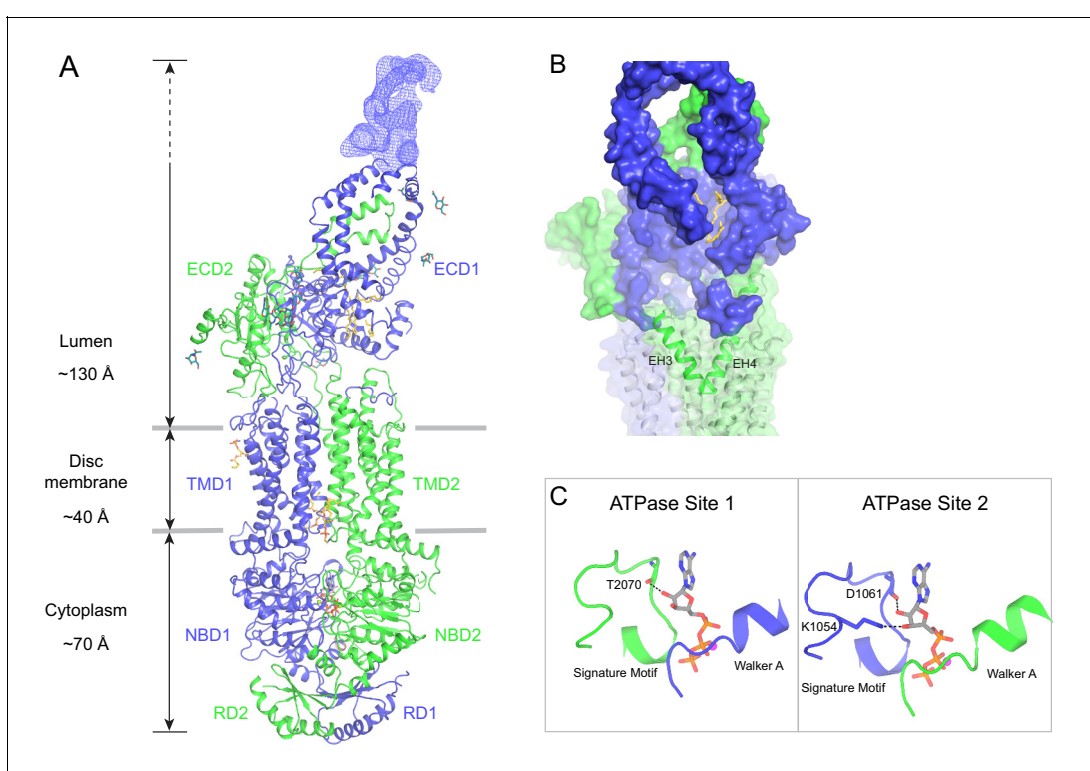

**Figure 5.** The structure of ABCA4 in the presence of ATP. (A) Ribbon diagram of the ATP-bound ABCA4 EQ mutant, together with electron microscopy density of the lid region. (B) Surface representation of the exocytoplasmic domains. The EH3-turn-EH4 motif is shown in green ribbon; lipids are shown as stick model. (C) Structure of the two ATPase sites. ATP is shown as sticks. Dashed lines indicate the non-consensus hydrogen bonds stabilizing the ribose moiety. EH: exocytoplasmic helices.

The online version of this article includes the following figure supplement(s) for figure 5:

**Figure supplement 1.** Cryo-electron microscopy reconstructions of the ATP-bound, EQ ABCA4.
**Figure supplement 2.** Local density of the ATP-bound, EQ ABCA4 reconstruction.

**Table 2.** Summary of electron microscopy data and structure refinement statistics for EQ ABCA4.

| *Data collection* | |
|---|---|
| Microscope | Titan Krios (FEI) |
| Voltage (kV) | 300 |
| Detector | K2 Summit (Gatan) |
| Pixel size (Å) | 1.03 |
| Defocus range (μm) | 0.8–2.5 |
| Movies | 9070 |
| Frames/movie | 50 |
| Dose rate (electrons/pixel/s) | 8 |
| Total dose (electrons/Å$^2$) | 75 |
| Number of particles | 1,136,006 |
| *Model composition* | |
| Non-hydrogen atoms | 15,513 |
| Protein residues | 1920 |
| Lipids | 8 |
| ATP | 2 |
| Mg$^{2+}$ | 2 |
| Sugar molecules | 14 |
| *Refinement* | |
| Resolution (Å) | 3.3 |
| Sharpening B-factor (Å$^2$) | −93.8 |
| Root-mean-square deviations | |
| Bond lengths (Å) | 0.007 |
| Bond angles (°) | 0.898 |
| *Validation* | |
| Molprobity score | 2.10 |
| Clashscore, all atoms | 12.42 |
| Favored rotamers (%) | 97.99 |
| Ramachandran plot (%) | |
| Favored | 91.68 |
| Allowed | 8.32 |
| Outliers | 0.0 |

In the presence of ATP and absence of substrate, ABCA4 exhibits a conformation very different from that of the ATP-free structure (*Figure 5*). The two NBDs form a closed dimer. The two TMDs are also in close contact, leaving no cavity at their interface (*Figure 5A*). The ECDs are positioned differently from that of the ATP-free form, placing the hollow channel directly above the EH3-turn-EH4 cleft (*Figure 5B*).

The ATP-binding signature motifs of ABCA4 deviate from the consensus LSGGQ sequence. The signature motif of NBD1 is LSGGM and that of NBD2 is YSGGN. In a typical ATPase site, Q in the LSGGQ signature motif positions the ribose moiety of ATP through an H-bond and van der Waals' contact (*Davidson and Chen, 2004*). In ABCA4, this role is fulfilled by the mainchain carboxyl group of T2070 in ATPase site 1 and the side chain of K1054 and mainchain carboxyl of D1061 in ATPase site 2 (*Figure 5C*). Consequently, the two ATP molecules are bound in a manner typical of ABC transporters.

ABCA4 binds and hydrolyzes GTP as effectively as ATP (*Ahn et al., 2000*; *Illing et al., 1997*). The structure indicates that both ATPase sites can bind GTP without steric hindrance. As rod

photoreceptor cells contain similar levels of ATP and GTP (*Biernbaum and Bownds, 1985*), these observations suggest that ABCA4 might utilize either nucleotide to power retinal translocation.

## Nature of the conformational changes

The large conformational changes of ABCA4 are initiated in the NBDs where two ATP molecules bind and stabilize a closed NBD dimer. Using the RD dimer as a reference, the two NBDs move like a pair of tweezers akin to that of the maltose transporter (*Oldham et al., 2007*; *Figure 6A, B*). At the base of the tweezers, two short helices pin the two NBDs together from opposing sides of the molecule (*Figure 6A, B*, cylinders). One of the pinning helices (PHs) is formed by residues following RD1 and the other PH contains the C-terminal, highly conserved, VFVNFA motif (*Fitzgerald et al., 2004*; *Patel et al., 2019*). In the absence of ATP, each PH simultaneously interacts with the H loop of one NBD and the D loop of the other NBD (*Figure 6A*). Upon ATP-binding, the NBDs rotate toward each other, forming additional contacts with the RDs and PHs that are functionally important (*Figure 6B*). For example, in the ATP-bound conformation, D1102 in NBD1 forms a hydrogen bond with H2202 in RD2, and R2107 in NBD2 forms salt bridges with E1270 in PH1 (*Figure 6B*). Mutations D1102Y and R2107H, which disrupt these interactions, are observed in patients with vision diseases (*Fritsche et al., 2012*; *Zernant et al., 2014*). Notably, the R2107H mutation is the most frequent mutation found in African Americans (*Zernant et al., 2014*).

How does NBD dimerization in the cytosol affect the TMDs to alter access of the translocation pathway? In opposing contrast to typical ABC transporters that are inward-facing when their NBDs are separated, the resting state of ABCA4 is outward-facing (*Figure 2*). Thus, the nature of the conformational change in ABCA4 is qualitatively different from the 'rocker-switch' motion described for many transporters (*Abramson et al., 2003*; *Huang et al., 2003*). Transition from the ATP-free to ATP-bound conformation involves two motions of the TMDs. The first is a movement in concert with the NBDs toward the molecular center, bringing the TMDs closer to each other. The second is a 30° twisting motion of TMD1 relative to the RecA-like subdomain (*Figure 6C*). The helical subdomain of NBD1 moves together with TMD1. This twisting motion, not seen in any other ABC transporter, places helices TM1 and TM2 within van der Waals' contact with TM8 and TM11, closing the TM cavity completely (*Figure 6D*).

## Discussion

Although a large body of clinical data indicates that ABCA4 plays a vital role in vision, we are still at an early stage in understanding its molecular mechanism. In this work, we present the structures of ABCA4 in two different conformations. These structures provide a molecular basis to understand why many clinically relevant mutations (*Figure 3—figure supplement 1*) could lead to misfolding or malfunction of ABCA4. They also reveal several intriguing features of ABCA4 that could be important to its function. For example, the ECDs contain an extended lipid-binding groove that changes its position in the presence of ATP. The EH3-turn-EH4 cleft, unique in ABCA4, seems well-positioned to lower the barrier for lipid headgroups to flip across the membrane. The ACT-fold of the RDs suggests that they may interact with small molecules to regulate the activity of ABCA4. Taken together, the structural information provides a molecular basis to design new experiments to uncover how ABCA4 recruits, flips, and releases retinoids and how these processes are regulated in the vision cycle.

The two structures of ABCA4 support an alternating access model for retinal import (*Figure 7*). In the resting state, ABCA4 adopts an outward-facing conformation, poised to recruit substrate from the exocytoplasmic side of the membrane. ATP binding induces a major conformational change that closes the transmembrane pathway to release substrate. ATP hydrolysis resets the transporter to begin a new cycle. Several lines of evidence support this working model. In the absence of ATP, N-retinylidene-PE and all-*trans* retinal bind ABCA4 with micromolar affinity (*Beharry et al., 2004*; *Zhong and Molday, 2010*) consistent with substrate recruitment to the outward-facing resting state. The addition of ATP, but not ADP, releases substrate (*Beharry et al., 2004*; *Zhong and Molday, 2010*), likely due to the closure of the TM cavity in the NBD-dimerized conformation (*Figure 6*). Many details of the ABCA4 transport cycle, such as whether an inward-facing conformation exists and how substrate is recruited and released, remain to be determined.

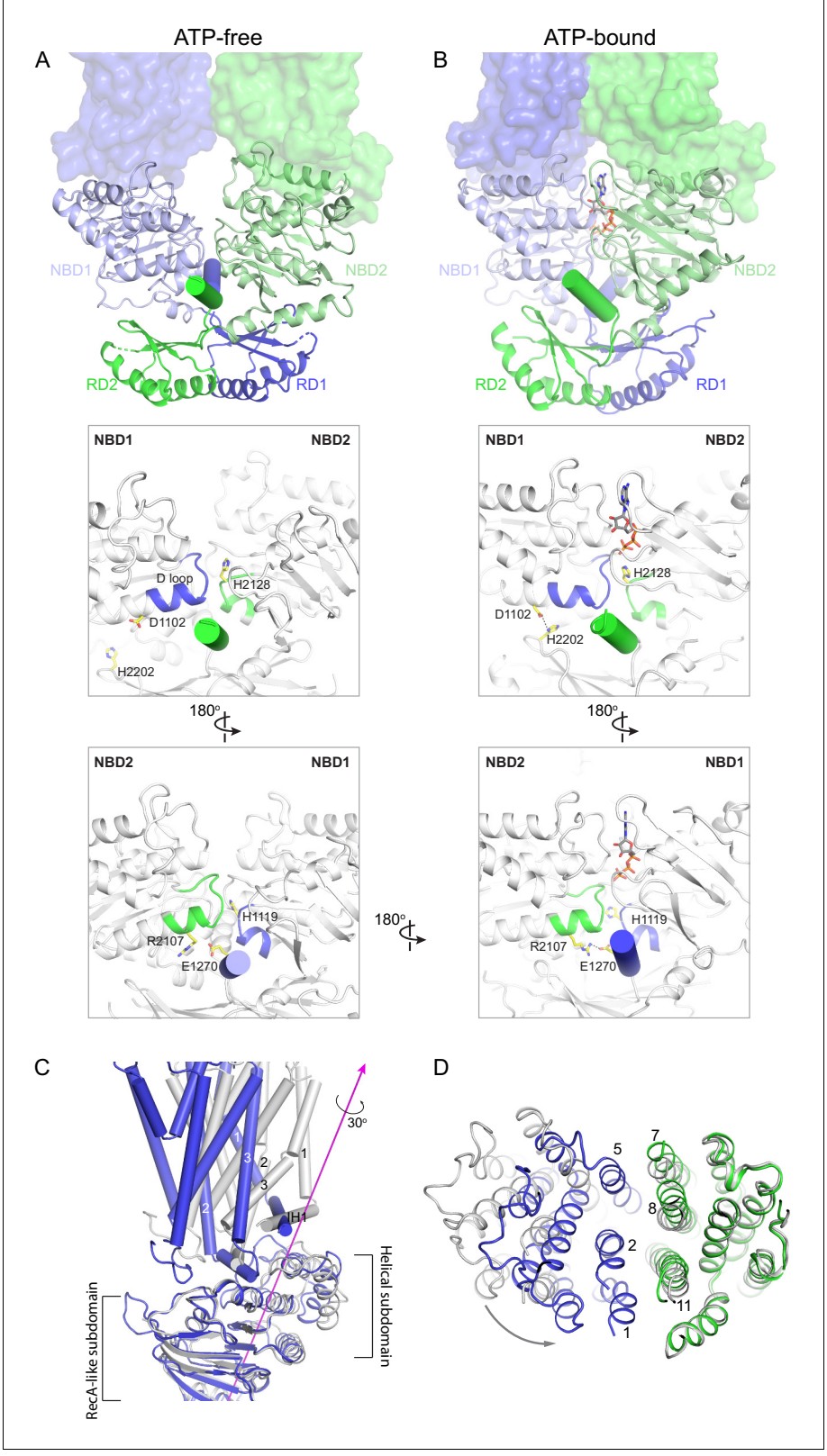

**Figure 6.** Conformational changes upon ATP binding. (**A**, **B**) The tweezer-like motion of the nucleotide binding domains (NBDs). Transmembrane domains (TMDs) are shown as surface representations; NBDs are displayed as ribbons. The two pinning helices are highlighted as cylinders. ATPs are shown as sticks. (**C**) The twisting motion of TMD1. The two structures are superpositioned based on NBD1. The ATP-bound form is shown in blue and ATP-

*Figure 6 continued*

free conformation in gray. The rotation axis of TMD1 and the helical subdomain is indicated in magenta. (**D**) Closing of the TM cavity upon TMD1 twisting. The structures are aligned based on TMD2. ATP-bound TMDs are shown in blue and green; ATP-free conformation is shown in gray.

As more ABC transporter structures become available, it is now evident that one cannot differentiate importers from exporters based on their structures. For example, the previously defined 'type I exporter fold' has now been observed in importers YbtPQ (*Wang et al., 2020*), IrtAB (*Arnold et al., 2020*), and ABCD4 (*Xu et al., 2019*). The structures of ABCA1 and ABCA4 are also very similar (*Figure 7—figure supplement 1*). Their TMDs are essentially superimposable, and the ECDs also share a common fold. The two structures differ significantly in the cytoplasmic region: ABCA1 does not exhibit the domain-swapped configuration as observed in ABCA4 (*Figure 7—figure supplement 1*); however, this difference is unlikely to be the determinant of their functional difference. The transport directionality of ABCA1 and ABCA4 is likely to be governed by when the substrates are recruited and released.

# Materials and methods

## Cell culture

Sf9 cells were cultured in Sf-900 II SFM medium (GIBCO) 5% FBS and 1% antibiotic-antimycotic. HEK293S GnTI-cells were cultured in Freestyle 293 (GIBCO) supplemented with 2% FBS and 1% antibiotic-antimycotic.

## Mutagenesis

Mutations (E1087Q, E2096Q) were introduced using QuikChange Site-Directed Mutagenesis System (Stratagene).

## Protein expression and purification

DNA encoding the human ABCA4 gene was synthesized and codon-optimized for expression in mammalian cells (BioBasic) and subcloned into a BacMam expression vector (*Goehring et al., 2014*) with a C-terminal green fluorescent protein (GFP) tag. The resulting plasmid was transformed to DH10Bac *Escherichia coli* cells to produce bacmid DNA. Recombinant baculoviruses were first generated in Sf9 cells using Cellfectin II reagents (Invitrogen). The resulting P1 viruses were amplified for two more generations. To express the recombinant proteins, P3 virus (10% v/v) was used to infect

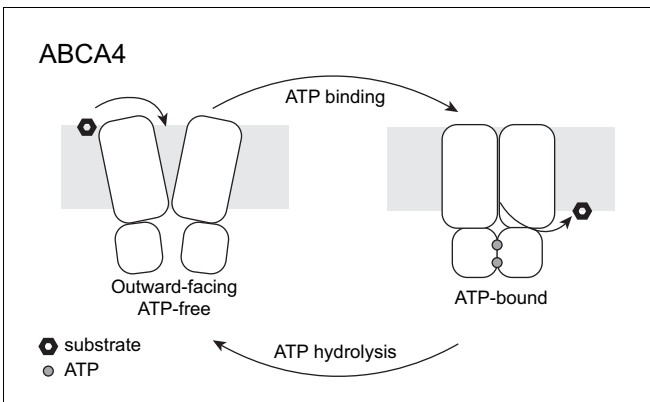

**Figure 7.** A working transport model. For simplicity, the extracellular domains and the substrate-binding proteins are omitted from the schematic drawings.

The online version of this article includes the following figure supplement(s) for figure 7:

**Figure supplement 1.** Structural comparison of ABCA4 and ABCA1.

HEK293S GnTI⁻ suspension cells at $3 \times 10^6$ cells/mL. Infected cells were cultured at 37°C for 12 hr before the temperature was decreased to 30°C. Protein expression was induced by adding 10 mM sodium butyrate at 30°C (*Goehring et al., 2014*). To purify the protein, cells were harvested after 48 hr post induction, resuspended in lysis buffer (50 mM HEPES pH 8, 2 mM MgCl₂, 200 mM NaCl, 20% Glycerol), and supplemented with protease inhibitors (1 μg/mL leupepetin, 1 μg/mL pepstatin, 1 μg/mL aprotonin, 100 μg/mL trypsin inhibitor, 1 mM benzamidine, and 1 mM phenylmethylsulfonyl fluoride [PMSF]) and DNase (3 μg/mL). Membranes were solubilized with 1.25% (w/v) 2,2-didecylpropane-1,3-bis-β-D-maltopyranoside and 0.25% (w/v) cholesteryl hemisuccinate for 2 hr at 4°C. The cell lysates were centrifuged at 70,000 *g* for 1 hr, and the supernatant was applied to CNBR-activated Sepharose resin (GE Healthcare) conjugated with anti-GFP nanobodies (*Kirchhofer et al., 2010*). The resin was washed with buffer A (20 mM HEPES pH 8, 200 mM NaCl, 2 mM MgCl₂, 0.06% digitonin) and then incubated with PreScission protease (5:1 w/w ratio) at 4°C for 3 hr to remove the C-terminal GFP tag. The protein was eluted with buffer A and further purified with gel filtration chromatography using a Superose 6 10/300 column (GE Healthcare) equilibrated with buffer A. The EQ ABCA4 sample was purified similarly except that 1 mM ATP was supplemented to the lysis buffer and buffer A.

## EM data acquisition and processing

Protein eluted from the gel filtration column was concentrated to 6 mg/mL. The EQ ABCA4 sample was further incubated with ATP and MgCl₂ (10 mM final concentration) on ice for 15 min. About 3 mM (final concentration) fluorinated Fos-choline-8 was added to samples right before freezing on Quantifoil R1.2/1.3 400 mesh Au grids using Vitrobot Mark IV (FEI). EM images were collected on a 300 kV Titian Krios (FEI) with a K2 Summit detector (Gatan) in super-resolution mode using Serial EM. The defocus ranged from 0.8 to 2.5 μm, and the dose rate was 8 e-/pixel/s.

For WT ABCA4, 5226 images were collected. The movie frames were first corrected for gain reference and binned by 2 to obtain a physical pixel size of 1.03 Å. Beam-induced sample motion was corrected using MotionCor2 (*Zheng et al., 2017*). CTF estimation was performed using Gctf (*Zhang, 2016*). In total, 1,375,284 particles were picked automatically using Gautomatch (http://www.mrc-lmb.cam.ac.uk/kzhang) and imported in cryoSPARC v2 (*Punjani et al., 2017*) for 2D classification. After 2D classification, 425,348 particles were selected for ab initio reconstruction in cryoSPARC. The best class (320,102 particles) was chosen for non-uniform (NU) refinement to yield a map of 3.64 Å. The half maps, masks, and particles from cryoSPARC were imported to RELION 3 (*Scheres, 2012*; *Zivanov et al., 2018*) through pyem suite (https://github.com/asarnow/pyem; *Asarnow et al., 2019*) for postprocessing and polishing. The polished particles were then imported to cryoSPARC for another round of NU refinement to yield a final map of 3.27 Å.

The EQ ABCA4 datasets were processed similarly with the following differences, and the best map was obtained by combining two separate datasets. For the first dataset, 6766 images were collected and 857,074 particles were picked by Gautomatch. 2D classification yielded 455,456 particles for ab initio reconstruction, and the best class (287,299 particles) was chosen for NU refinement to obtain a 3.57 Å map. Further postprocessing and polishing in RELION 3 yielded a map of 3.37 Å. For the second dataset, 2304 images were collected and 278,932 particles were picked. 2D classification yielded 121,894 particles for ab initio reconstruction, and the best class (46,432 particles) was chosen for NU refinement to obtain a 3.83 Å map. Further postprocessing and polishing in RELION 3 yielded a map of 3.52 Å. The best classes from both datasets (287,299 particles and 46,432 particles) were combined after polishing and imported into cryoSPARC to obtain a final map of 3.27 Å with NU refinement.

## Model building, refinement, and analysis

Model building and refinement were carried out as previously described (*Zhang and Chen, 2016*). In brief, each dataset was randomly split into two halves, one half for model building and refinement and the other half for validation. Iterative model building and real space refinement were performed in Coot (*Emsley et al., 2010*) and Phenix (*Afonine et al., 2018*; *Liebschner et al., 2019*), respectively. The final model of WT ABCA4 includes 1941 residues: 3–117, 118–137 (polyalanine), 272–468, 479–496 (polyalanine), 497–873, 916–937, 947–1162, 1201–1277, 1340–1900, 1911–2172, and 2178–2253; 14 lipids, one detergent (digitonin), and 14 sugar molecules (10 2-acetamido-2-deoxy-beta-D-

glycopyranose molecules and 4 beta-D-mannopyranose molecules). The final model of EQ ABCA4 includes 1920 residues: 3–117, 118–137 (polyalanine), 273–279 (polyalanine), 280–303, 308–331, 339–350, 362–448, 449–457 (polyalanine), 458, 459–468 (polyalanine), 498–502 (polyalanine), 503–881, 914–1164, 1199–1280, 1342–1902, 1916–2173, 2178–2252, 2 $Mg^{2+}$, 2 ATPs, 8 lipids, and 14 sugar molecules. Structural model validation was done using Phenix and MolProbity (*Chen et al., 2010*). The local resolutions were estimated using RELION 3 (*Scheres, 2012*; *Zivanov et al., 2018*) by using half maps from cryoSPARC (*Punjani et al., 2017*).

Domain movements were analyzed with Dyndom (http://dyndom.cmp.uea.ac.uk/dyndom/). Figures were generated with PyMOL (Schrödinger, LLC) and UCSF Chimera (*Pettersen et al., 2004*).

## ATPase assay

The basal ATPase activity (*Figure 1C*) was measured using an ATP/Nicotinamide adenine dinucleotide (NADH) consuming coupled method (*Scharschmidt et al., 1979*) in reaction buffer (50 mM HEPES pH 8.0, 150 mM KCl, 0.06% digitonin, 2 mM $MgCl_2$, 60 μg/mL pyruvate kinase, 32 μg/mL lactate dehydrogenase, 9 mM phosphoenolpyruvate, 0.15 mM NADH). WT ABCA4 or EQ ABCA4 were diluted to 0.2 μM in reaction buffer. A control buffer was also prepared without ABCA4. Different concentrations of ATPs were added to initiate the reaction, and the fluorescence changes of NADH were recorded by a Tecan Infinite M1000 microplate reader at excitation wavelength of 340 nm and emission wavelength of 445 nm. To quantify the ATPase activity, mean values and standard deviation from three independent measurements were calculated. The values for $K_m$ and the specific turnover rates were determined by nonlinear regression of the Michaelis–Menten equation using GraphPad Prism 8. The maximal ATPase activities were calculated assuming a molecular weight of 256 kDa for human ABCA4.

Retinal stimulation of ATPase activity of ABCA4 (*Figure 1D*) was measured using a Transcreener ADP2 FI Assay (Bellbrook Labs) in kinetic mode according to the manufacturer's recommendations. Reactions were assembled in triplicate in black, round-bottom, 384-well plates (Corning #5414) under dim red light. Also, 10 μL reactions were prepared by incubating ABCA4 with varying concentrations of all-*trans* retinal or all-*trans* retinoic acid for 15 min on ice, then adding ATP and 10 μL of ADP2 detection mix containing tracer and antibody. The final reaction mixture contains 50 nM ABCA4, 100 μM ATP, 20 mM HEPES, 75 mM NaCl, 3 mM $MgCl_2$, 1 mM Dithiothreitol (DTT), 0.005% glyco-diosgenin (GDN), 0.001% Brij-35, 1% ethanol, and the concentration of substrate as indicated. Fluorescence changes were recorded by a Tecan Infinite M1000 microplate reader at excitation wavelength of 580 nm and emission wavelength of 620 nm. Standard curves mimicking the conversion of ATP to ADP were used to convert raw fluorescence measurements to ADP formation. The data were analyzed by GraphPad Prism 8.

## Acknowledgements

We thank Mark Ebrahim and Johanna Sotiris at The Rockefeller Evelyn Gruss Lipper Cryo-Electron Microscopy Resource Center for assistance in data collection. JL is a fellow of the Helen Hay Whitney Foundation. We also thank The Rockefeller University and the Howard Hughes Medical Institute for financial support. The authors declare no competing financial interests.

## Additional information

### Funding

| Funder | Author |
| --- | --- |
| Howard Hughes Medical Institute | Jue Chen |
| Helen Hay Whitney Foundation | James Lee |

The funders had no role in study design, data collection and interpretation, or the decision to submit the work for publication.

## Author contributions
Fangyu Liu, Conceptualization, Data curation, Investigation, Writing - original draft, Writing - review and editing; James Lee, Data curation, Writing - review and editing; Jue Chen, Conceptualization, Supervision, Funding acquisition, Investigation, Writing - original draft, Writing - review and editing

## Author ORCIDs
Fangyu Liu (iD) https://orcid.org/0000-0001-5022-0106
James Lee (iD) https://orcid.org/0000-0002-8551-0258
Jue Chen (iD) https://orcid.org/0000-0003-2075-4283

## Decision letter and Author response
Decision letter https://doi.org/10.7554/eLife.63524.sa1
Author response https://doi.org/10.7554/eLife.63524.sa2

## Additional files

### Supplementary files
• Transparent reporting form

### Data availability
The cryo-EM maps are deposited in the Electron Microscopy Data Bank (EMDB) under accession codes: EMD-23409, EMD-23410. The corresponding atomic models are deposited in the Protein Data Bank (PDB) under accession codes 7LKP and 7LKZ.

The following datasets were generated:

| Author(s) | Year | Dataset title | Dataset URL | Database and Identifier |
|---|---|---|---|---|
| Liu F, Lee J, Chen J | 2021 | Structure of ATP-free human ABCA4 | https://www.ebi.ac.uk/pdbe/entry/emdb/EMD-23409 | Electron Microscopy Data Bank, EMD-23409 |
| Liu F, Lee J, Chen J | 2021 | Structure of ATP-bound human ABCA4 | https://www.ebi.ac.uk/pdbe/entry/emdb/EMD-23410 | Electron Microscopy Data Bank, EMD-23410 |
| Liu F, Lee J, Chen J | 2021 | Structure of ATP-free human ABCA4 | https://www.rcsb.org/structure/7LKP | RCSB Protein Data Bank, 7LKP |
| Liu F, Lee J, Chen J | 2021 | Structure of ATP-bound human ABCA4 | https://www.rcsb.org/structure/7LKZ | RCSB Protein Data Bank, 7LKZ |

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
