## [Decision Letter]

**Acceptance summary:**

Jue Chen and co-workers present two cryo-EM structures of the retinal importer ABCA4 in the apo and ATP-bound states. ABCA4 is an ABC importer with exporter fold, which transports retinal and toxic retinoids from the disc lumen to the cytosol. Overall, the cryo EM structures of the only known ABC importer in human, is an important accomplishment and reveals an interesting architecture with potentially novel mechanistic and physiological ramifications. This study further adds to the growing body of evidence that one cannot differentiate importers from exporters based on their structures.

**Decision letter after peer review:**

Thank you for submitting your article "Molecular structures of the eukaryotic retinal importer ABCA4" for consideration by *eLife*. Your article has been reviewed by three peer reviewers, including David Drew as the Reviewing Editor and Reviewer #1, and the evaluation has been overseen by Olga Boudker as the Senior Editor. The following individual involved in review of your submission has agreed to reveal their identity: Konstantinos Beis (Reviewer #3).

The reviewers have discussed the reviews with one another and the Reviewing Editor has drafted this decision to help you prepare a revised submission.

Summary:

Fangyu Liu and Jue Chen present two cryo-EM structures of the retinal importer ABCA4 in the apo and ATP-bound states. ABCA4 is an ABC importer with exporter fold, which transports retinal and toxic retinoids from the disc lumen to the cytosol. Overall, the cryo EM structures of the only known ABC importer in human is an important accomplishment and reveals an interesting architecture with potentially novel mechanistic and physiological ramifications.

Essential revisions:

While the reviewers appreciated the beautiful structural biology and well-written manuscript, we feel the current paper falls short on the functional characterization of the protein and a comparison of the structural features of ABCA4 with that of other ABC transporters, in particular ABCA1, which is an lipid exporter.

1) We think it is important to demonstrate that the preparation used for structural work is able to carry out substrate-dependent ATPase activity as expected I.e., stimulation of ATPase activity with N-retinylidene-PE and/or 11-cis-retinal or the all-trans isomers. Moreover, can stimulation of ATPase activity by substrate be discriminated from possible trans-inhibitory effects? e.g. by titrating the protein with a wide range of substrate concentrations

2) The proposed substrate binding site(s) were unclear. A paper was cited that concluded that retinal binds to the ECD domains, but it was unclear if retinal was retained in the current preparation, which should be able to be monitored spectroscopically. Please clarify. Furthermore, is retinal thought to be transported in addition to N-retinylidene-PE or this an allosteric site? If not, how would the substrate binding site in the ECD domain be coupled to the TMD interface? Is the thickness of the rim disc membrane bilayer known? If the idea is that lipid-like substrates are first recruited to the ECD (as implied by analogy to importers and retinal binding site in the ECDs), wouldn't these domains be too far from the membrane bilayer or could the lipid bilayer adjust to reach up to the ECD domains? We would recommend ConSurf analysis to show sequence conservation overlaid on the ABCA4 structure to provide support for the substrate recognition site(s) and import pathway. Also, please clarify the type of lipids modelled in TMD interface and how these were chosen, i.e., was the purified protein analyzed for bound lipids? Furthermore, please clarify if these lipids thought to represent the N-retinylidene-PE binding-site location?

3) We would like to see structural comparisons of ABCA4 to ABCA1 that would help to illuminate the structural interpretation leading to the proposed import mechanism. This is particularly important since ABCA1 appears to have the same overall fold and is an exporter, and not an importer. We are conscious a paper focusing on the general biology of ABCA4 vs. the "expert audience", however we are of the opinion that in balance this analysis would only strengthen the current paper overall.

– What is the relative positing of the TMDs in ABCA1 vs. ABCA4?

– In ABCA1 the TMDs are positioned far apart and this positioning has been suggested to be a detergent artefact. However, ABCA4 the TMDs are similarity positioned, which would provide support that these weak, open TMD interfaces are physiologically relevant. Indeed, the presence of lipids between the TMDs in ABCA1 suggests this arrangement could be a means to recruit lipid substrates to the most likely substrate-binding site. Is there any clear structural difference between ABCA1 and ABCA4 that would indicate their preference for different lipid substrates at the TMD interface?

– Is there any structural differences that would make ABCA1 a lipid exporter but ABCA4 a lipid importer?

– How does the ECDs from ABCA1 structurally compare to those from ABCA4? For instance, are the hydrophobic grooves in the ECD domain in ABCA1 similarity positioned to their location in ABCA4?

4) Points 1. and 2. would be clearly strengthened by specific ABCA4 mutants that were shown to either impacted substrate(s) binding and/or that could be used to probe the preferred orientation of ABCA4, i.e., it was unclear how it was concluded that the preferred conformation is outward-facing? Depending on the ability to provide further experimental support for the proposed substrate binding site(s) and the import pathway, we think the final mechanistic model should be appropriately adjusted, i.e., toned-down if necessary.

---

## [Author Response]

Essential revisions:While the reviewers appreciated the beautiful structural biology and well-written manuscript, we feel the current paper falls short on the functional characterization of the protein and a comparison of the structural features of ABCA4 with that of other ABC transporters, in particular ABCA1, which is an lipid exporter.1) We think it is important to demonstrate that the preparation used for structural work is able to carry out substrate-dependent ATPase activity as expected I.e., stimulation of ATPase activity with N-retinylidene-PE and/or 11-cis-retinal or the all-trans isomers. Moreover, can stimulation of ATPase activity by substrate be discriminated from possible trans-inhibitory effects? e.g. by titrating the protein with a wide range of substrate concentrations

Because retinoids absorb strongly at 445 nm, the ATP/NADH-coupled assay cannot be used to measure the ATPase activity in the presence of substrate. Therefore, we had to establish a new assay, the Transcreener ADP2 FI Assay (Bellbrook Labs), to measure the ATPase activity as a function of substrate concentration. In the revision, we added a new Figure 1D showing that, consistent with literature, retinal stimulates the ATPase activity only in the presence of PE.

2) The proposed substrate binding site(s) were unclear. A paper was cited that concluded that retinal binds to the ECD domains, but it was unclear if retinal was retained in the current preparation, which should be able to be monitored spectroscopically. Please clarify.

There is a misunderstanding here. We did not mean to propose a specific substrate binding site. We have reworded the text to avoid such confusion.

To test if retinal was retained in the current preparation, the absorbances of the retinal and protein samples were measured over a range of wavelengths. It is clear that our preparations (both WT and EQ) do not contain any retinal.

Furthermore, is retinal thought to be transported in addition to N-retinylidene-PE or this an allosteric site?

Retinal is thought to be transported in the form of N-retinylidene-PE, as transport of retinal requires PE (Quazi et al., 2012) and ABCA4 preferentially binds N-retinylidene-PE compared to free retinal (Beharry et al., 2004). We stated in the text: “ABCA4 enables this process by moving all-*trans* retinal into the cytoplasm, likely in the form of its phosphatidylethanolamine (PE) conjugates, N-retinylidene-PE.”

If not, how would the substrate binding site in the ECD domain be coupled to the TMD interface? Is the thickness of the rim disc membrane bilayer known? If the idea is that lipid-like substrates are first recruited to the ECD (as implied by analogy to importers and retinal binding site in the ECDs), wouldn't these domains be too far from the membrane bilayer or could the lipid bilayer adjust to reach up to the ECD domains? We would recommend ConSurf analysis to show sequence conservation overlaid on the ABCA4 structure to provide support for the substrate recognition site(s) and import pathway.

If the ECD functions to recruit the substrate, one can imagine it has to undergo large conformational change to deliver the substrate to the TMDs. Although there isn’t sufficient evidence to support this hypothesis, we still would like to discuss this possibility to stimulate future studies. We reworded the text to be clear:

“Prokaryotic ABC importers often contain an extracytoplasmic binding protein that functions as a receptor for substrates (Davidson et al., 2008). […] Previously it was shown that isolated ECD2 binds to all-*trans* retinal (Biswas-Fiss et al., 2010), but how this interaction relates to the transport cycle remains unclear.”

Also, please clarify the type of lipids modelled in TMD interface and how these were chosen, i.e., was the purified protein analyzed for bound lipids? Furthermore, please clarify if these lipids thought to represent the N-retinylidene-PE binding-site location?

We built lipid acyl chains, cholesterol, and digitonin based on the shape of the electron density. We don’t think these lipids represent the N-retinylidene PE binding site. Instead, we suggest that “lipids form an integral part of the retinoid transport system, possibly by regulating the folding and function of ABCA4”.

3) We would like to see structural comparisons of ABCA4 to ABCA1 that would help to illuminate the structural interpretation leading to the proposed import mechanism. This is particularly important since ABCA1 appears to have the same overall fold and is an exporter, and not an importer. We are conscious a paper focusing on the general biology of ABCA4 vs. the "expert audience", however we are of the opinion that in balance this analysis would only strengthen the current paper overall.

As requested, we added a figure comparing the structures of ABCA4 with ABCA1 in their apo forms (Figure 7—figure supplement 1). The structure of ATP-bound ABCA1 has not been published.

– What is the relative positing of the TMDs in ABCA1 vs. ABCA4?

In the apo form, the TMDs of ABCA1 and ABCA4 are positioned very similarly ( Figure 7—figure supplement 1).

– In ABCA1 the TMDs are positioned far apart and this positioning has been suggested to be a detergent artefact. However, ABCA4 the TMDs are similarity positioned, which would provide support that these weak, open TMD interfaces are physiologically relevant. Indeed, the presence of lipids between the TMDs in ABCA1 suggests this arrangement could be a means to recruit lipid substrates to the most likely substrate-binding site. Is there any clear structural difference between ABCA1 and ABCA4 that would indicate their preference for different lipid substrates at the TMD interface?

The TMDs of ABCA4 and ABCA1 are very similar. We do not think the lipids observed in the structure of ABCA4 represent the potential substrate binding site. Instead, we stated that:

“The gap between the TMDs at the level of the inner leaflet is filled with ordered lipids (Figures 4A and B). […] These observations indicate that lipids form an integral part of the retinoid transport system, possibly by regulating the folding and function of ABCA4”.

– Is there any structural differences that would make ABCA1 a lipid exporter but ABCA4 a lipid importer?

There isn’t a clear structural difference between ABCA1 and ABCA4 to correlate with their functions. We have added the following discussion to the manuscript:

“As more ABC transporter structures become available, it is now evident that one cannot differentiate importers from exporters based on their structures. […] The transport directionality of ABCA1 and ABCA4 is likely to be governed by when the substrates are recruited and released”.

– How does the ECDs from ABCA1 structurally compare to those from ABCA4? For instance, are the hydrophobic grooves in the ECD domain in ABCA1 similarity positioned to their location in ABCA4?

The hydrophobic groove of ABCA4 is larger than that of ABCA1 (Figure 7—figure supplement 1). Both are similarly positioned in the ATP-free form (Figure 7—figure supplement 1).

4) Points 1. and 2. would be clearly strengthened by specific ABCA4 mutants that were shown to either impacted substrate(s) binding and/or that could be used to probe the preferred orientation of ABCA4, i.e., it was unclear how it was concluded that the preferred conformation is outward-facing? Depending on the ability to provide further experimental support for the proposed substrate binding site(s) and the import pathway, we think the final mechanistic model should be appropriately adjusted, i.e., toned-down if necessary.

Unfortunately, no mutant of ABCA4 has been characterized to impact substrate binding. As suggested, we have toned down the presentation of the model (Discussion).